# Activity and Affinity of Pin1 Variants

**DOI:** 10.3390/molecules25010036

**Published:** 2019-12-20

**Authors:** Alexandra Born, Morkos A. Henen, Beat Vögeli

**Affiliations:** 1Department of Biochemistry and Molecular Genetics, University of Colorado Anschutz Medical Campus, 12801 East 17th Avenue, Aurora, CO 80045, USA; alexandra.born@cuanschutz.edu (A.B.); morkos.henen@cuanschutz.edu (M.A.H.); 2Faculty of Pharmacy, Mansoura University, Mansoura 35516, Egypt

**Keywords:** pin1, WW domain, PPIase domain, mutants, activity, affinity

## Abstract

Pin1 is a peptidyl-prolyl isomerase responsible for isomerizing phosphorylated S/T-P motifs. Pin1 has two domains that each have a distinct ligand binding site, but only its PPIase domain has catalytic activity. Vast evidence supports interdomain allostery of Pin1, with binding of a ligand to its regulatory WW domain impacting activity in the PPIase domain. Many diverse studies have made mutations in Pin1 in order to elucidate interactions that are responsible for ligand binding, isomerase activity, and interdomain allostery. Here, we summarize these mutations and their impact on Pin1′s structure and function.

## 1. Introduction

Pin1, or Protein interacting with Never-in-Mitosis (NIMA) 1, is a phosphorylation-dependent peptidyl-prolyl *cis-trans* isomerase (PPIase) [1]. Unlike the other PPIases, the cyclophilins and FK506 binding proteins, Pin1 is a two-domain protein specific for isomerizing motifs with a proline immediately preceded by a phosphorylated serine or threonine (pS/T-P) [2]. The N-terminal WW (Trp-Trp) domain binds these motifs, while the C-terminal PPIase domain is responsible for isomerization [3]. Connecting the two domains is a 10 residue flexible linker (Figure 1A). Interestingly, the relative positions between the two domains are not fixed, as the two domains can occupy “compact” and “extended” states. In the compact state, the two domains interact through an interdomain (ID) interface between residues 28–32 in the WW domain, and 137–141 and 148–149 in the PPIase (Figure 1A) [4,5]. The position of the domains in the extended state is expected to be a distribution of states, rather than just one [6,7,8]. An equilibrium exists between compact and extended conformations of Pin1 that can be altered by ligands [9,10]. Ligand binding and mutation studies suggest that the WW domain allosterically regulates the activity of the PPIase domain.

Pin1 is a very promiscuous mitotic regulator, and regulates phosphoproteins through changing the phosphorylation/dephosphorylation state, as well as protein stability through either enhancing or protecting against ubiquitin-mediated proteasomal degradation [11]. Pin1 is overexpressed in many different cancer types, including lung, brain, melanoma, prostate, ovary, and cervical [12]. In breast cancer, Pin1 has been implicated in RAS/MEK/ERK, WNT/β-catenin, NFκB, HER2, and ERα signaling pathways [13,14,15,16,17]. More specifically, Pin1 enhances proteosomal degradation of c-Myc and Cyclin E, while protecting CDK1, β-catenin, NFκB, and p53. Pin1 has even been shown to promote cancer stem cell metastasis and tumorigenesis [18]. While overexpression of Pin1 is implicated in cancer, Pin1 typically protects against tauopathy and plaque formation that leads to Alzheimer’s disease (AD) neurodegeneration. Pin1 has been shown to regulate both Tau and Aβ, that lead to intracellular tangles and extracellular plaques, respectively [19,20,21,22]. It is apparent that targeting Pin1 for therapeutics is a substantial challenge, especially due to its opposite roles in AD and cancer.

It is clearly apparent that Pin1 is capable of binding and isomerizing many different substrates, all with the commonality of a pS/T-P motif (Figure 1). Despite this shared motif, not all substrates cause the same structural changes to Pin1 upon binding. A peptide derivative from the pT48-P49 site of M phase inducer phosphatase Cdc25C with sequence EQPLpTPVTDL (here called pCDC25c) causes Pin1 to favor a hyper-extended state compared to the apo condition [8]. On the other hand, peptides with sequences WFYpSPR, FFpSPR, and YpSPTpSPS cause a shift to a more compact state of Pin1 [6]. This ligand-dependent shift in interdomain equilibrium triggers different responses in PPIase dynamics and catalytic activity. The Pintide ligand with sequence WFYpSPR was discovered as the peptide that Pin1 has the highest activity to isomerize [3], and ligand FFpSPR is similar in sequence but is slightly more hydrophilic and therefore has higher solubility in aqueous buffers. Ligand YpSPTpSPS is based on the repeated sequence in the C-terminal domain of RNA Polymerase II. The primary ligands that have been examined in this review include pCDC25c, FFpSPR, and the CTD of RNA Pol II with peptides drawn in Figure 1B.

Pin1 has been very extensively studied for the last 20 years, with most structural work aimed at understanding the dynamics and allostery of this two-domain protein. Various studies have been performed looking at Pin1 binding to many different ligands. In this review, we focus specifically on a summary of which mutants in Pin1 impact catalytic activity and ligand binding affinity and to what extent. As such, this review highlights mutations made in biochemical and structural studies. We hope this review aids in computational studies and methodological developments in order to characterize an allosteric system on a per residue basis.

The association of variants with specific diseases is outside the scope of this review. However, the interested reader is referred to an excellent review on single nucleotide polymorphisms and mutations of Pin1 in cancers [23]. Many other articles exist discussing the role of Pin1 in cancer and Alzheimer’s disease, for example references [20,21,24,25,26,27,28]. Other reviews describe the role of Pin1 in transcription [29], the cell cycle and signaling [30,31,32], and its role as a molecular switch in proteasomal degradation [11]. Lastly, a comprehensive review of the structure and function of Pin1 summarizes the current knowledge in the field [33].

## 2. Mutation Impact on Affinity to Various Ligands

### 2.1. Methods for Determination of Affinity and Activity

Appendix A lists many Pin1 variant dissociation constants with the ligand of interaction, the method used, the buffer conditions, and the study that the variant was reported in. Techniques that have been used to evaluate Pin1 binding include fluorescence anisotropy, isothermal titration calorimetry (ITC), and nuclear magnetic resonance (NMR) titrations. As Pin1 is a two domain protein with two distinct binding sites, NMR is the most sensitive method for determining binding since the two binding sites can be evaluated simultaneously and residue specifically (microscopic binding affinities). The main drawback for NMR is that it is ideal only for weaker binding events in the μM to mM regime. Fluorescence anisotropy and ITC have the potential for showing two separate binding events, but the *K_D_* of the two events must be different by orders of magnitude, as if they are too similar they are indistinguishable. Even if two binding events were measured using either of these two methods, these methods alone would not be enough to disentangle which binding affinity corresponds to each binding site. In all the studies that have reported binding affinities of Pin1 using fluorescence anisotropy or ITC, only one binding affinity was reported for the whole protein, likely because the binding affinities of each domain are in the same order of magnitude, resulting in an effective *K_D_*. The smaller the *K_D_* value, the larger the binding affinity between the two molecules.

To quantify Pin1 isomerase activity, most mutants were studied using NMR exchange spectroscopy (EXSY) by fitting the *cis-trans* and *trans*-*cis* cross and the respective diagonal peaks in a NOESY experiment using many different mixing times [34]. Before performing this EXSY experiment, the ^1^H chemical shifts of ligand are assigned without enzyme present. The ^1^H chemical shifts are often distinct for *trans* and *cis* species due to slow exchange, typically with peak intensity ratios around 10:1 *trans* to *cis*. During a NOESY experiment, usually used to measure through-space interactions, each hydrogen is frequency labeled by its respective chemical shift and after a certain mixing time (time given for isomerization to occur), the hydrogens are frequency labeled again. If the proton isomerized to a different chemical shift during that time, there will be a cross peak between the two distinct frequencies. For example, if a ligand is in the *trans* conformation during the first frequency detection and then undergoes isomerization during the mixing time, the *cis* conformation is detected at the second frequency labeling event. Isomerization of a ligand can occur without enzyme present, but this is often a slow process (seconds-minutes) that is unlikely to occur during a NOESY mixing time (milliseconds-seconds). For pCDC25c and FFpSPR ligands, no cross peaks are detected without enzyme present. Once Pin1 is added, cross-peak intensities can be measured at various mixing times to extract rates of exchange. A perk of this experiment is that both *c*→*t* and *t*→*c* rates can be measured, and the sum of which is the exchange rate (*k*_CT_ + *k*_TC_ = *k*_EXSY_). While many studies report both *k*_CT_ and *k*_TC_, some studies only report *k*_EXSY_. Therefore, in the bulk of this report we look at *k*_EXSY_ values for ease of comparison, but *k*_CT_ and *k*_TC_ values are reported in the Appendix A for many Pin1 variants. Note that the larger the *k*_EXSY_ value, the higher is the activity of the enzyme.

Another method to measure activity by determining *k*_cat_/*K*_M_ has been performed through a chymotrypsin-coupled chromophoric assay of p-nitroaniline (pNA) [35]. Chymotrypsin is highly selective towards X-Pro-Phe-pNA and will only hydrolyze the C-terminal pNA if the X-Pro bond is in the *trans* conformer. Therefore, the substrate used to study Pin1 activity is suc-AEPF-pNA, with glutamate acting as a phosphomimetic. A similar assay can be done with trypsin when peptides contain Pro-Arg-pNA motifs, like with peptide WFYpSPR-pNA [3]. Appendix A reports the activity for studies using this method. Here, the larger the *k*_cat_/*K*_M_, the higher is the activity of the enzyme.

Figure 2 and Figure 3 summarize the affinity and activity, respectively, of many Pin1 mutants listed in the Appendix A. Single measurements are reported only with a bar, while variants with multiple measurements show an error bar with the standard deviations. All values (with their individual experimental errors) are reported in the Appendix A. Note that some values have a high standard deviation do to various measurements not agreeing well across different studies.

### 2.2. Experimental Mutant-Dependent Affinities and Activities

Both domains have been studied in isolation, where they have different binding affinities and activities than the full-length (FL) protein (Figure 2A). It should be noted that constructs that are tested in the full-length protein always have “FL” written, while the isolated domains do not have any distinction. For example, “WW” refers to the isolated WW domain, while “FL WW” is the WW domain in full-length Pin1. This nomenclature is carried out throughout this review. While the isolated WW domain has similar affinity to the WW domain in full-length Pin1 (albeit marginally weaker), the isolated PPIase has typically 100× weaker affinity than the PPIase in the full-length protein [10]. It should be noted the high variability in PPIase binding: values range from 8 μM [5] to 10 mM [36]. Using GST pull-downs, the isolated WW domain was able to pull-down mitotic phospho-proteins (containing MPM-2 and Plk1 antibody recognition motifs) with similar efficiency as full-length Pin1, while the isolated PPIase domain had no ability to pull-down these targets [37]. Overall, we can conclude that the PPIase has lower binding affinity than the WW domain to the tested ligands and that the presence of the partnering domain has an effect on the affinity of each domain. All mutants evaluated in this study are located on the structure of Pin1 in Figure 4.

All reported values for binding affinities and activities refer to phosphorylated ligands. It has been shown, however, that Pin1 can also bind unphosphorylated motifs. For example, the activity of Pin1 towards unphosphorylated WFYSPR-pNA is 170 mM^−1^s^−1^, it increases to 20,160 mM^−1^s^−1^ for WFYpSPR-pNA (see Appendix A) [3]. The initial crystal structure, 1pin, contained a bound phosphate ion in the catalytic site [4]. Interestingly, other PPIases such as the cyclophilins and FKBPs are unable to isomerize phosphorylated motifs.

#### 2.2.1. Mutations in the WW Domain

The N-terminal WW domain is comprised of residues 1–39, and consists of a 3-stranded β-sheet (Figure 1A). While the WW domain is unable to isomerize prolines, it can bind pS/T-P motifs in ligands. The contribution to stability of the entire isolated WW domain of each residue has been tested by complete alanine-scanning and partial glycine-scanning mutagenesis, and only four residues resulted in an unfolded β-sheet: W11A, Y24A, N26A, and P37A [38]. Random mutagenesis of full-length Pin1 showed that all but 6 residues of the WW domain were able to be mutated in yeast [39]. The hypomutable residues identified are G10, W11, R21, N30, A31, and S32. Interestingly, residues 30–32 are near the interdomain interface of the WW domain. In addition, the sequence alignment of Pin1 identified only W11 and S32 as completely conserved in different species [39]. Residue W11 is vital for structural stability, while S32 is likely important for ligand binding (which likely explains why S32 could be mutated to an alanine in the previous stability study [38]). Activity and affinity were not evaluated on these mutants.

A fluorescence anisotropy binding study of full-length Pin1 was performed using ligand with sequence YpSPTpSPS, which is the repeated region of the C-terminal domain of RNA Pol II [40]. In this analysis, residues in the WW domain binding site were mutated in order to investigate the energetics of this interface (Figure 2B). Mutating residues 16, 17, 23 (in Loop 1) and 34 cause a large reduction in the 10 μM binding affinity of the WW domain (Figure 4), and are therefore likely more important to binding ligands than the β-sheet itself. Another report confirmed that S16A, Y23A, Y23F, and W34A resulted in a loss of function in yeast [41]. The β-sheet has been tested by mutating residues 14 and 25, which results in similar or even slightly higher affinities, depending on the amino acid type chosen for mutation. Residues S16, R17, and Y23 are known to be important for binding the phosphate group of the ligand, and provide neutral hydrogen bonds. It is likely that mutation S16H has a larger negative impact on the affinity than S16A due to the introduction of a cation in physiological conditions as well as the addition of a bulkier, aromatic side chain, which could also sterically hinder the ligand interaction. In addition, a study on the isolated WW domain reported that deleting residue S19 not only shortens loop 1, but this change reduces the binding affinity and changes the dynamics of this loop by reducing the loop flexibility [42]. This loss of loop 1 flexibility is speculated to impair the malleability of the phosphate binding pocket. Y23F causes a negative impact on the affinity as the hydroxyl group on the tyrosine is important for coordinating a water molecule that stabilizes the negatively-charged phosphate group as well. Residue W34 is one of the namesake tryptophans in this domain, and it is clear that the large aromatic side-chain is critical for ligand binding, as even an aromatic reduction to a phenylalanine negatively impacts WW binding affinity. It should be noted that in this study involving binding to the CTD of Pol II, and activity was not assayed for these mutants [40].

A study was performed to investigate the structure-folding relationship of the WW domain specifically looking at the WW binding loop (residues 17–20) as it is the rate limiting step of folding for this domain [43]. In this study, mutations were inserted and truncations of this loop were performed, with the WT WW domain with residues ^17^RSSG^20^, and mutants ^17^-ADG^20^, ^17^-RDG^20^, ^17^-ARG^20^, ^17^-SSG^20^, ^17^RSS-^20^, ^17^--NG^20^, ^17^--RG^20^, and ^17^--SG^20^. Interestingly, all these loop mutants resulted in increased stability upon thermal denaturation. The majority of these loop mutants also fold faster than WT Pin1 WW. Conversely, upon measuring the affinity of WW loop variants on binding to the single phosphorylated CTD of RNA Pol II (sequence YSPTpSPS), only WT Pin1 WW had any measurable affinity using isothermal titration calorimetry with a *K_D_* of 35 ± 25 μM. Of the other variants that were measured, none showed any measurable binding and heat transfer via ITC. The authors conclude that the WT WW domain is tuned for phospho-peptide binding at the expense of increased folding time and stability [43].

Full-length mutant W34A was investigated in other studies with ligand pCDC25c, and was found to have reduced activity compared to WT Pin1 [44]. W34 appears to be significant for ligand binding the WW domain, and this binding to the WW domain may just simply increase the local concentration of the ligand to the catalytic site, so with perturbed binding, the activity appears to be reduced. Alternatively, it is possible that due to the interdomain allostery, this mutation interrupts the link between the ligand binding site on the WW domain and the PPIase catalytic site. Mutant Q33E of the isolated WW domain was also investigated due to its unexpected role in thermostability of the domain [45]. A glutamine to glutamate mutation results in the substitution of a hydrogen donator for a hydrogen acceptor and of a neutral for a negative charge. Despite this amino acid on the periphery of the WW domain and not directly involved in ligand binding, this mutation results in a decrease in pCDC25c binding compared to the isolated WW domain. This Q33E mutation may be disrupting the hydrophobic core holding together the WW domain.

Overall, it can be concluded that structural integrity of the WW β-sheet is well tolerated for most mutations except for W11, Y24, N26 and P37. Mutating and/or truncating the WW binding loop (residues 17–20) and mutating S32 and W34 severely impacts substrate binding.

#### 2.2.2. Mutations in the PPIase Domain

Unlike the mutations studied in the WW domain involving full-length Pin1, mutations evaluated in the PPIase domain were from a mixture of studies with either the isolated PPIase domain or full-length Pin1. For the analysis of each mutation, it is noted whether this mutation was studied in the isolated or full-length construct. The C-terminal PPIase domain is residues 50–163, and contains a β-sheet core with surface-exposed α-helices (Figure 1A). The PPIase catalytic site is mostly structured, but there is a semi-disordered catalytic loop (residues 63–80) residing above the catalytic site. Based on the original crystal structure of Pin1, the peptidyl-prolyl isomerase catalytic site consists of residues H59, C113, S154, and H157 [4].

Unigenic evolution using random mutagenesis in yeast showed three hypomutable regions in the PPIase domain: R56-P70, A107-G120, and F139-I158 [39]. These three regions are all near the catalytic site. It was found that residues K63, R68, and R69 of the catalytic loop are vital for phosphate binding in the pS/T-P motif. Mutating residues R68 and R69 in the PPIase domain does not impact the WW domain’s binding affinity to ligand pCDC25c in full-length Pin1 (Figure 2C), but this mutation completely inhibits PPIase activity (Figure 3A), as no detectable exchange was observed with pCDC25c [44]. In addition, the R68A/R69A mutant Pin1 was unable to pull-down mitotic phospho-proteins in HeLa lysate [37]. Using the chymotrypsin degradation assay, mutants R68A/R69A and K63A result in a large reduction in catalytic activity [39] (Figure 3C).

Cysteine 113 has been noted as a key residue for the activity of Pin1, yet a C113D mutation still maintains some isomerase activity in the isolated PPIase domain [36]. Cysteines are prone to oxidation in the cell, and an aspartate mutation can be considered an oxidative mimetic. Pin1 has previously been found to be oxidized in the hippocampus of Alzheimer’s disease samples, and the activity in this oxidized Pin1 is reduced [24]. Oxidation of residue C113 may be the link to reduced Pin1 activity in AD. This mutation mainly impacts residues nearby in the catalytic site and disturbs the hydrogen bond network formed with histidines 59 and 157, and this structural change also alters the basic triad (K63, R68, R69) responsible for binding the phosphate group. But, the overall fold of the catalytic site is maintained with this mutation. C113A and C113S mutations were also tested. Interestingly, while a serine is the best mimetic for cysteine for having a similar size side-chain and being a hydrogen bond donor, no activity was actually detected of this mutant unlike aspartate and alanine [46]. This C113 study was performed on the isolated PPIase domain, so there is currently no evidence on if or how this mutation impacts the WW domain. While R68A/R69A mutations resulted in the inability of Pin1 to bind to various targets in HeLa lysate, the C113S mutation in Pin1 was able to maintain its target binding function [37]. This study suggests that C113 is responsible for isomerase activity yet has little impact on target binding.

While it is clear that residue C113 participates directly in catalysis, the role of residues H59 and H157 in the active site is less obvious. The original hypothesis was that these histidines stabilize the transition state of the ligand [4]. Initial yeast viability assays showed that H59 mutated to bulky amino acids (L, F) resulted in reduced growth, while smaller side chains (A, N, S) maintained viability [47]. All H157 mutants (A, N, S, L, and F) supported viability. Interestingly, a double mutation of H59L and H157L/A/F completely rescued the viability previously impaired by H59L alone, suggesting that two hydrophobic residues at these positions support viability. Activities of these mutants were measured on a trypsin degradation assay with Pintide-pNA as the substrate (Figure 3D). The H59L mutation results in less than 5% activity of WT Pin1, while the double mutation H59L/H157L increases the efficiency to ~15% of WT activity. By comparing the activity of mutations to the viability of these mutations in yeast, it appears that 5% of WT activity is necessary to maintain cell viability. This study also reported activity of many other mutations at H57 and H159, as shown in Figure 3D. Overall, this report suggests that the primary role of H59 and H157 is to maintain structural integrity of the active site, rather than being directly involved in catalysis.

The tumor suppressor death-associated protein kinase 1 (DAPK1) has been shown to phosphorylate S71. This phosphorylation event reduces the catalytic activity of Pin1 [48] as residue 71 is located in the catalytic loop responsible for binding substrate in the PPIase domain. Since DAPK is a tumor suppressor, in tumor cells that have a reduction in DAPK, DAPK is not present to tightly regulate Pin1′s activity, leading to hyperactive Pin1. Full-length mutant S71E (glutamate as a phosphomimetic) was shown to have reduced isomerase activity compared to WT Pin1 [49]. The introduction of negative charge to this region via phosphate/glutamate may disrupt the positive side chains K63 and R69 that make up the “basic triad” that bind the phosphate group of the ligand. This mutation causes structural and electrostatic perturbations not only in the catalytic loop, but also in the PPIase domain core. Further work in this study looked at residue D112 which is at a “critical juncture” with residues K63, S72, R74, and R80 that forms critical contacts through hydrogen bonding and salt bridges in various deposited Pin1 structures. These hydrogen bond “latches” were then investigated to see if there was a connection between the catalytic loop (res ~63–80) and the loop connecting helices 2 and 3 (res 111–114) via mutation studies. Mutants R74A, D112A, and D112N all caused a huge reduction in the catalytic activity, supporting the importance of this hydrogen bonding network. Unexpectedly and still without complete explanation, R80Q actually caused an increase in PPIase activity compared to WT even with a side-chain charge flip (cation to anion), suggesting that the electrostatics are not completely understood in Pin1.

The combination of mutation studies show that residue C113 is important for catalysis, residues H59 and H157 are vital for the structure of the catalytic site, and K63, R68, and R69 are responsible for phospho-substrate binding.

#### 2.2.3. Mutations in the Interdomain Interface and Linker

The two domains of Pin1 are connected via a flexible linker (residues 40–49), and there is an interdomain interface where the two domains are transiently associated. The interdomain interface is comprised of residues 28–31 of the WW domain, and 137–142 of the PPIase domain. It has been suggested that this interface is responsible for the allosteric connection between the two ligand binding sites [10].

The most well characterized mutant in the ID interface is I28A. Mutating this residue causes chemical shift perturbations in the ID interface of the PPIase domains (residues 137–141), and decreases ID contact between the two domains [5,8]. This I28A mutation also causes altered dynamics in the backbone and the side-chains of the PPIase domain [5]. Interestingly, this mutant had lower affinity for the ligand pCDC25c in the WW domain, but its isomerase activity was actually higher than WT [5,10]. This residue is neither in the WW binding site nor in the catalytic site, which supports the theory that the two domains are allosterically coupled.

Due to the higher activities of the isolated PPIase domain and the I28A mutant compared to WT Pin1, there was a hypothesis that interdomain contact negatively regulates catalysis since I28A causes a further extended state and the isolated PPIase domain has no interdomain contact [5,8]. This hypothesis was also supported by pCDC25c binding that causes an increase of the extended state of Pin1 compared to no ligand present. Conversely, the ligand FFpSPR has been shown to increase the interdomain contact, and there had been no conclusive evidence to prove that this lowered PPIase activity [50]. Therefore, the interdomain linker was modified to see if this impacted interdomain contact and activity while keeping the two structured domains intact. The ten-residue linker is residues 40-NSSSGGKNGQG-50. Many linker mutations were created with various linker length, but only two constructs were analyzed for activity with different number of residues deleted: 41ΔSG and 40ΔNSSSG [9]. It should be noted that the linker lengths were determined and tested based on linker lengths of Pin1 found in other species. 40ΔNSSSG adopted the most compact conformation out of all the mutants that were studied, but both constructs ended up having higher catalytic activity than WT Pin1. This strongly suggests that there is no direct correlation between interdomain contact and PPIase activity. To ensure that the increase in activity was due solely to the increase in interdomain contact and not due to WW binding, W34A/R17A mutant, which shows no binding in the WW domain at all, was also investigated. W34A/R17A had reduced isomerase activity compared to WT Pin1. When 40ΔNSSSG was mutated in with W34A/R17A, activity increased compared to the W34A/R17A control. It is surprising how the 40ΔNSSSG/W34A/R17A mutant actually had slightly higher activity (*k*_EXSY_ = 74.8 s^−1^) than 40ΔNSSSG (*k*_EXSY_ = 66 s^−1^), but this discrepancy was not addressed [9].

Another mutation investigated on the isolated PPIase domain was S138A, which is proposed to mimic the structural impact of the WW domain ID contact. S138 is located in the ID interface of the PPIase domain. Pin1 is phosphorylated at residue S138 by mixed-lineage kinase 3 (MLK3) which results in an increase in catalytic activity of Pin1. An *in vivo* study compared the activity of WT, S138E (phospho-mimetic), and S138A (phospho-deficient) Pin1 and showed that S138E had 4 times greater catalytic activity than WT Pin1 in a cellular assay, and the S138A mutant had 2-fold decrease in activity [51]. The S138E-PPIase construct had low expression in *E. coli* and was extremely unstable at room temperature, therefore it was unsuitable for NMR studies. These challenges were not present for S138A-PPIase mutant [46]. The S138A mutant caused minimal structural changes, but significantly altered the dynamics in the catalytic site. Using [15] N-HSQC, NOE, and H/D exchange spectra, it is apparent that the catalytic site is slightly reorganized and the hydrogen-bonding network has been altered due to the S138A mutation. The S138A mutation in the isolated PPIase may be mimicking the impact of the presence of the WW domain [46]. This would help it explain why it has lower catalytic activity than the WT isolated PPIase.

The I28A and S138A mutations disrupt the interdomain allostery, which results in higher isomerase activity. The linker length between the two domains does not appear to have a clear correlation to the isomerase activity.

#### 2.2.4. Impact on *cis*- and *trans*-Locked Inhibitors on Pin1

An in-depth study of non-isomerizable ligands was investigated on full-length Pin1, as well as its isolated domains. Ligand FFpSPR has a very similar sequence to Pintide, WFpSPR, but has increased solubility. Chemists were able to “lock” the proline in either a *cis* or *trans* state by replacing the peptidyl-prolyl linkage by alkene isosteres [52]. These non-isomerizable ligands are inhibitors to Pin1′s activity, as the alkene restricts isomerization. Interestingly, while the PPIase domain is able to bind both the *cis*- and *trans*-locked inhibitors, the WW is specific for the *trans* [50]. In addition, the chemical shift perturbations in the PPIase domain are distinct for the *cis*- and *trans*-locked inhibitors, suggesting that the binding modes for the two states are different. As the PPIase domain has a preference to bind *cis*-locked FFpSPR (lower *K_D_*), when *trans*-FFpSPR is first titrated in, which binds to both domains, subsequent addition of *cis*-FFpSPR displaces the *trans* state from the PPIase domain, and the *cis* binding signatures become apparent. The PPIase has about 10 times higher affinity for the *cis*-inhibitor than the *trans*-inhibitor (7 μM versus 66 μM), and this could be a contributing factor to why the exchange rate between *cis*-*trans* (*k*_CT_) is typically 10 times higher than *trans*-*cis* (*k*_TC_), with WT Pin1 usually having values reported about *k*_CT_ = 40 s^−1^ and *k*_TC_ = 4 s^−1^ [9]. Even upon protein mutation, the activity difference between these two isomerization rates remains (See Appendix A). 

#### 2.2.5. Differences in Allosteric Responses Due to pCDC25c Versus FFpSPR Binding

There is a plethora of evidence suggesting that ligands pCDC25c (sequence EQPLpTPVTDL) and FFpSPR have different allosteric pathways in Pin1, with the former stabilizing the extended state and the latter stabilizing the compact state [10]. Interestingly, these ligands have different effects on the dissociation constants and activity of Pin1 and its isolated domains as well. Both ligands have higher affinity for the WW domain than the PPIase domain in full-length Pin1, plus negligible binding (mM affinity to no binding detected) in the isolated PPIase domain. While pCDC25c has similar affinity for the FL WW domain as well as the isolated WW domain (*K_D_* ~ 6–10 μM), FFpSPR has a much greater affinity in the FL WW (*K_D_* ~ 43 μM) compared to the isolated WW (*K_D_* ~ 246 μM). This suggests that that PPIase domain has a positive impact on the WW domain’s ability to bind FFpSPR, while the PPIase domain doesn’t have an effect when the WW binds to pCDC25c. This is potentially due to the fact that FFpSPR stabilizes the compact state (increased interdomain contact) and this contact allosterically changes the WW binding site to enhance binding. Since the extended state (decreased interdomain contact) is preferred for pCDC25c, there isn’t a mechanism for the PPIase to strengthen the affinity of the WW domain. While the isomerase activity of FL Pin1 and the isolated PPIase domain is somewhat variable depending on the study, we can confidently conclude that isomerization of pCDC25c is higher in the isolated PPIase than FL Pin1 (*k*_EXSY_ range of ~37–58 s^−1^ versus ~26–44 s^−1^). On the other hand, the *k*_EXSY_ of FFpSPR is between 44–97 s^−1^ for FL Pin1 while only a single measurement of 91 s^−1^ for the isolated PPIase. Therefore, no confident conclusions can be made in regard to FFpSPR effect on activity, but at first glance it appears it doesn’t have much of a change on activity compared to pCDC25c. Ligand pCDC25c causes a stabilized extended state with less interdomain contact, and the isolated PPIase doesn’t have any interdomain contact (since the WW is absent). A primary hypothesis is that the WW domain is a negative regulator of the PPIase activity in the compact state, and the extended state has higher isomerase activity (supported by the enhanced activity of mutant I28A which also causes a stabilized extended state compared to WT) [5]. Therefore, the isolated PPIase domain has higher activity than FL Pin1 on pCDC25c since the WW domain is not present to be able to negatively regulate it. But, this negatively allosteric regulation does not hold up when it comes to FFpSPR, which has been shown to have a stabilized compact state but no large, conclusive changes in activity between FL Pin1 and the isolated PPIase.

## 3. Conclusions

The changes in affinity and activity of Pin1 upon mutations are mapped on the Pin1 structure in Figure 4. There is no direct correlation between the effects on affinity and activity. Unsurprisingly with Pin1, like with most proteins, if the binding/active sites are mutated, in most cases a reduction in affinity and/or PPIase activity results (Figure 4). On the other hand, mutations at distal, strategic sites, like I28A and S138A in the interdomain interface, actually increase the isomerase activity through an allosteric mechanism (Figure 4B). In addition, changing the linker length between the two domains also changes the catalytic activity of Pin1. For ligand pCDC25c, the activity is higher for the isolated PPIase domain than in the presence of the WW domain. This questions the hypothesis that the purpose of the WW domain is the increase of local ligand concentration in order to increase the activity of Pin1. Instead, the WW domain appears to play a more subtle role in activity regulation, possibly in context with ligand-specific allostery.

Conclusively, this review summarized mutations previously studied in Pin1 with their impacts on affinity and activity, and we hope other researchers can use this to evaluate and develop models to describe allostery. A suitable model for allostery would have to be able to explain the extensively mapped out effects of the mutations. This overview could also serve as an aid to make further mutations. Some future projects could look at the double mutants I28A and A140I, as these two residues have been predicted as a co-evolving pair and could give further insights into interdomain allostery [5]. Previous work suggests a conserved, hydrophobic conduit linking the interdomain interface to the PPIase catalytic site [53]. Residues I28, T29, A140, V150, T152, L60, L61, and V62 could be mutant targets for understanding this putative allosteric conduit. These mutations must trigger a coupled response of intra- and interdomain structural dynamics of Pin1. We are currently investigating such effects by a combination of nuclear magnetic resonance (NMR) and electron paramagnetic resonance (EPR) spectroscopy to characterize intra- and interdomain spatial sampling of Pin1, respectively.

## Figures and Tables

**Figure 1 molecules-25-00036-f001:**
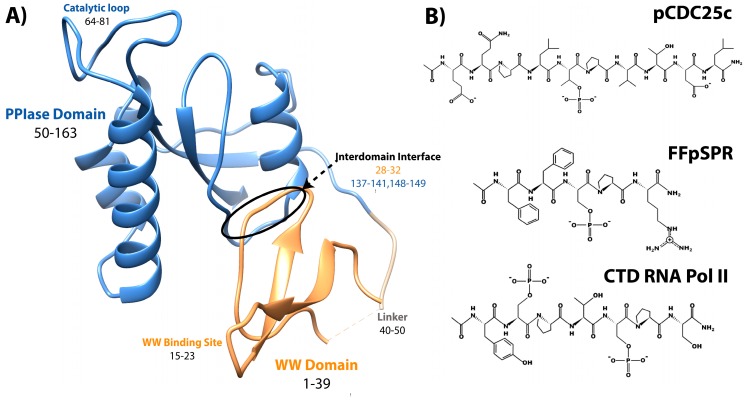
Structure of Pin1 and ligands. (**A**) Annotated crystal structure of PDB 1pin. (**B**) Primary pS/T-P peptide ligands discussed in this review.

**Figure 2 molecules-25-00036-f002:**
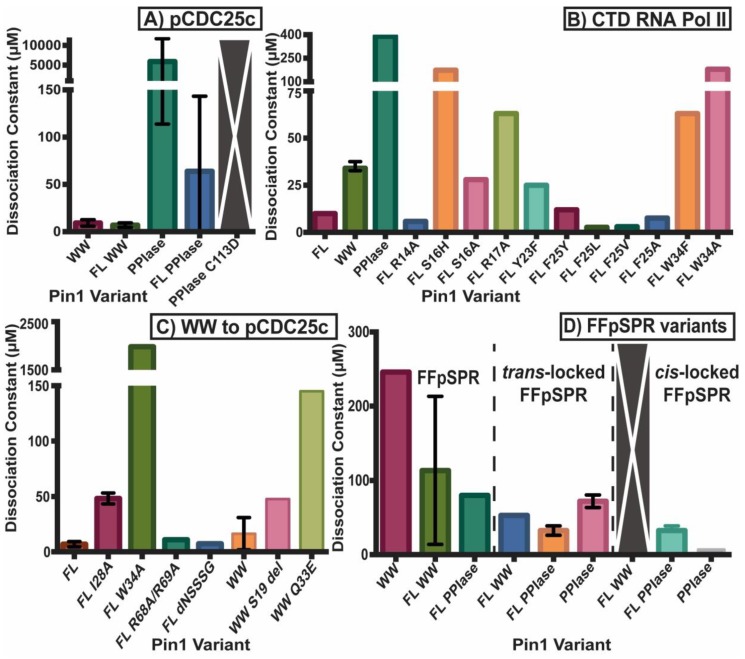
Dissociation constants of Pin1 variants. *K_D_* of (**A**) Pin1 variants including full-length (FL) and isolated domains to ligand pCDC25c, (**B**) mutants towards CTD RNA Pol II, (**C**) the WW domain and variants towards pCDC25c, and (**D**) full-length and isolated domains toward isomerizable and locked inhibitors of FFpSPR. Single measurements are reported only with a bar, while variants with multiple measurements show an error bar with the standard deviations. All values (with their individual experimental errors) are reported in the Appendix A. Variants with a large gray bar with an X had no detectable binding.

**Figure 3 molecules-25-00036-f003:**
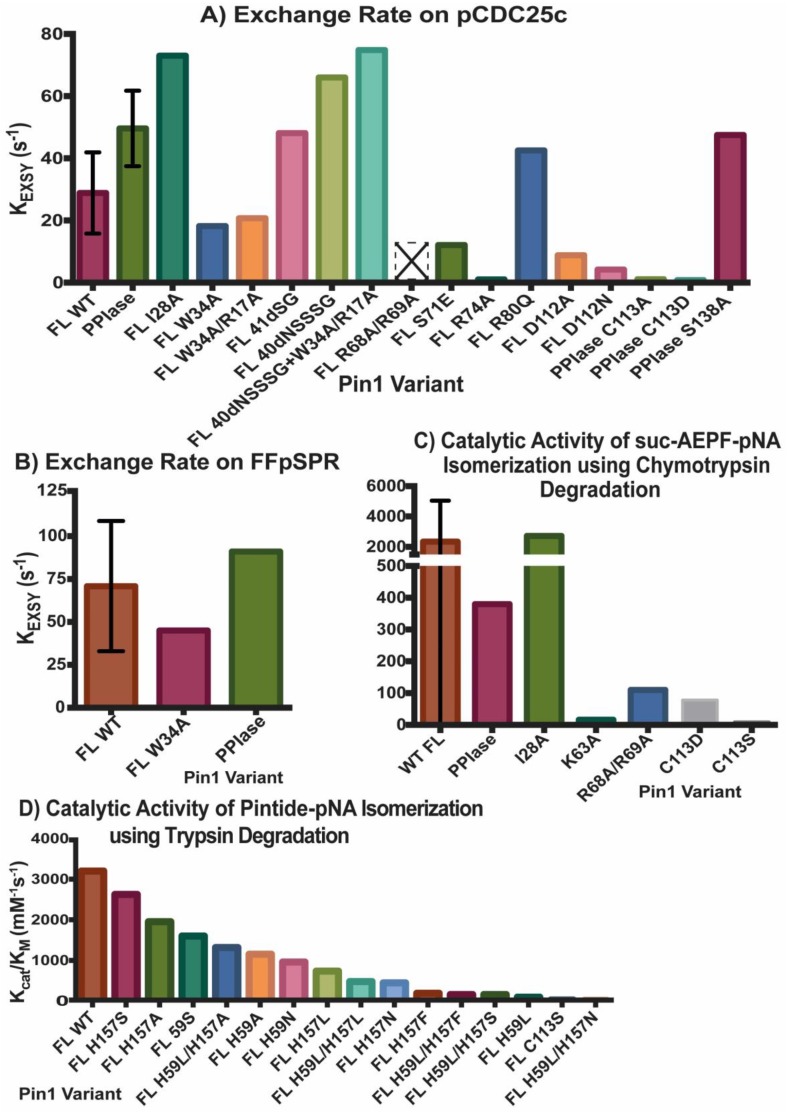
Activity of Pin1 Variants. Exchange rate *K*_EXSY_ of ligands (**A**) pCDC25c and (**B**) FFpSPR in presence of various Pin1 variants using exchange spectroscopy. Catalytic activity *k*_cat_/*K*_M_ of Pin1 mutations using (**C**) chymotrypsin or (**D**) trypsin assay. Single measurements are reported only with a bar, while variants with multiple measurements show an error bar with the standard deviations. All values (with their individual experimental errors) are reported in the Appendix A. Variant R68A/R69A with an empty bar with an X had no detectable activity. The values reported in (**D**) are based on the estimate of WT Pin1 activity (~3200 mM^−1^s^−1^) from the original figure, and the estimates of percent of activity for each mutant compared to WT Pin1. Appendix A also allows for 5% error in these values.

**Figure 4 molecules-25-00036-f004:**
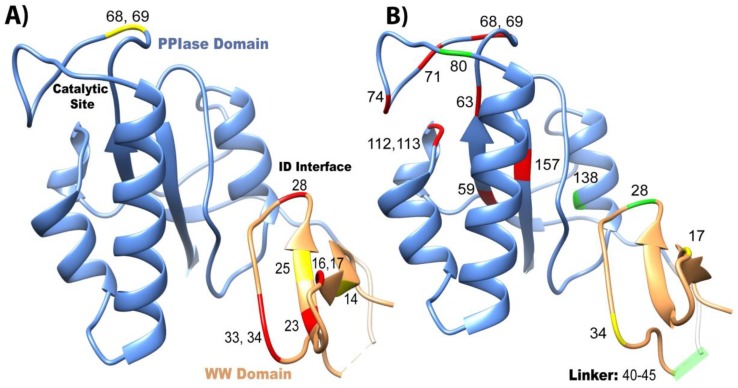
Overview of mutants that change affinity and activity of Pin1. Mutations that have been studied are plotted on the crystal structure 1pin. Mutations that negatively impact (**A**) affinity and (**B**) isomerase activity are labeled in red, positive mutations are in green, and neutral mutations are in yellow.

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
