# Peer review of "Activity and Affinity of Pin1 Variants"

_molecules, 2019, doi:10.3390/molecules25010036_

Round 1
Reviewer 1 Report
This review on Pin1 is timely and interesting, and will be very useful to researchers who want to learn about PPI-ase activity. Overall, the paper is well-written and engaging, although some minor language errors are present that could easily be fixed with a careful edit pass. The literature review is quite comprehensive and gives a good account of current progress in this field. The NMR experiments are described very well and the authors have done a good job of making the results understandable to those who are not NMR experts.
A few minor changes and additions could strengthen the manuscript. As this review is for a general bio-organic audience, it would be useful to make sure everyone understands the system up front, i.e. put the annotated protein structure (Fig. 3) at the very beginning. It would also be useful to add a figure showing a typical ligand. The biological context for the ligands is also uneven, with minimal explanation given for the ones discussed at the beginning of the paper, and more in-depth discussion given for the later ones. It would be helpful to have some biological context for all the major peptide ligands discussed here. Overall, this is a solid contribution.
Reviewer 2 Report
In this manuscript, the authors reviewed the activity and substrate affinity of various Pin1 mutants. The authors selected reports related to the activity and substrate affinity of Pin1 variants from a number of publications on Pin1 and summarized them in this review article. There are several reviews about Pin1 that have been published before. They focused on functional mechanisms, cellular functions, and disease-related functions. This review is valuable because it focuses on other fundamental aspects of Pin1 and its variants. However, this article does not include some important topics or publications. Some references are as follows; (1) Biochemistry, 2008, 47, 11481-11489 (dual Histidine motifs), (2) Proc. Natl. Accad. Sci. USA, 2006, 103, 10648-10653 (structure-function-folding relationship of WW domain), (3) J. Mol. Biol., 2007, 365, 1143-1162 (functionally important residues for PPIase activity), (4) Biochemistry, 2013, 52, 6968-6981 (interdomain interaction), (5) Front. Physiol., 2013, 4, 18 (substrate binding characteristics for both domains). These publications are just a few of the important publications. Therefore, this review is far from comprehensive. Therefore, we do not recommend posting this review article in this journal.
Reviewer 3 Report
The authors have reviewed the site-directed mutants of Pin1 that affect the affinity and activity in facing ligands or targets. The authors have provided information regarding the methods of determination of affinity and activity, mutations in the WW domain and the PPIase domain, and other areas. Without a doubt, any particular mutant exhibits an altered folding that affect its binding with ligands. While the review is comprehensive, the quality of the data shown in the figures 1 and 2 is largely compromised. The authors have failed to provide:
1) environmental conditions such as buffers, pH and serum that affect the binding between WW domain and PPIase domain, 2) the rationale of each mutant design and physiological relevance is unknown, 3) no perspectives are shown in the Discussion or a specific section, 4) the scenario of binding and catalysis of Pin1 with non-phosphorylated S/T-P motifs in the ligands and the effect of sites in the nearby or distant locations were not discussed, 5) structural comparison of the WW domain in Pin with other single WW domain protein would enhance the understanding of the underlying mechanism for protein/ligand interactions and the associated PPIase activity, 6) the association of Pin1 variants with specific diseases was not clearly specified, and 7) take home message is weak in each section. This would be critical in assisting the understanding of the article by readers not in the field.
Minor:
Figure legends for 2 and 3 were largely missing. Please pay attention to the final layout of your preprint for reviews to read and assess. Please clarify why DAPK1 limits Pin1 activity and the outcome of DAPK1 deficiency in cancer cells in affecting Pin1 in relevance to cancer cell growth and metastasis. Statistics in Figures 1 and 2 are unreliable. Not enough sample sizes were shown. This makes the presentation and discussion meaningless. In particular, the control data for full-length Pin1 in Figure 2 are too large in ranges. This is hard to believe. Similar scenario is shown in the Figure 1. Figure 3: Again, legend is missing. The figure can be depicted better to explain why each mutant acts in each distinct way.
Round 2
Reviewer 2 Report
The article has significantly improved. I am satisfied with the revise by the authors. This article is appropriate to be published in this journal.